# Improving Winter Wheat Photosynthesis, Nitrogen Use Efficiency, and Yield by Optimizing Nitrogen Fertilization

**DOI:** 10.3390/life12101478

**Published:** 2022-09-23

**Authors:** Muhammad Saleem Kubar, Khalid S. Alshallash, Muhammad Ahsan Asghar, Meichen Feng, Ali Raza, Chao Wang, Khansa Saleem, Abd Ullah, Wude Yang, Kashif Ali Kubar, Chenbo Yang, Samy Selim, Arafat Abdel Hamed Abdel Latef, Fatmah Ahmed Safhi, Salha Mesfer Alshamrani

**Affiliations:** 1College of Agriculture, Shanxi Agricultural University, Taigu 030801, China; 2College of Science and Humanities-Huraymila, Imam Mohammed Bin Saud Islamic University (IMSIU), Riyadh 11432, Saudi Arabia; 3Agricultural Institute, Center for Agricultural Research, ELKH, 2462 Martonvásár, Hungary; 4Chengdu Institute of Biology, Chinese Academy of Sciences (CAS), University of Chinese Academy of Sciences, Chengdu 610041, China; 5Department of Horticultural Sciences, The Islamia University of Bahawalpur, Bahawalpur 63100, Punjab, Pakistan; 6Xinjiang Key Laboratory of Desert Plant Roots Ecology and Vegetation Restoration, Xinjiang Institute of Ecology and Geography, Chinese Academy of Sciences, Urumqi 830011, China; 7Faculty of Agriculture, Lasbela University of Agriculture, Water and Marine Sciences, Uthal 90150, Balochistan, Pakistan; 8Department of Clinical Laboratory Sciences, College of Applied Medical Sciences, Jouf University, Sakaka 72388, Saudi Arabia; 9Botany and Microbiology Department, Faculty of Science, South Valley University, Qena 83523, Egypt; 10Department of Biology, College of Science, Princess Nourah bint Abdulrahman University, P.O. Box 84428, Riyadh 11671, Saudi Arabia; 11Department of Biology, College of Science, University of Jeddah, Jeddah 23218, Saudi Arabia

**Keywords:** winter wheat, nitrogen, photosynthesis, growth stages, grain yield

## Abstract

Wheat is the third most producing crop in China after maize and rice. In order to enhance the nitrogen use efficiency (NUE) and grain yield of winter wheat, a two-year field experiment was conducted to investigate the effect of different nitrogen ratios and doses at various development stages of winter wheat (*Triticum aestivum* L.). A total of five N doses (0, N75, N150, N225, and N300 kg ha^−1^) as main plots and two N ratios were applied in split doses (50%:50% and 60%:40%, referring to 50% at sowing time and 50% at jointing stage, 50% at sowing time + 50% at flowering stage, 50% at sowing time + 50% at grain filling stage, and 60% + 40% N ratio applied as a 60% at sowing time and 40% at jointing stage, 60% at sowing time and 40% at flowering stage, and 60% at sowing time and 40% at grain filling stage in subplots). The results of this study revealed that a nitrogen dose of 225 kg ha^−1^ significantly augmented the plant height by 27% and above ground biomass (ABG) by 24% at the grain filling stage, and the leaf area was enhanced by 149% at the flowering stage under 60 + 40% ratios. Furthermore, the N225 kg ha^−1^ significantly prompted the photosynthetic rate by 47% at the jointing and flowering stages followed by grain filling stage compared to the control. The correlation analysis exhibited the positive relationship between nitrogen uptake and nitrogen content, chlorophyll, and dry biomass, revealing that NUE enhanced and ultimately increased the winter wheat yield. In conclusion, our results depicted that optimizing the nitrogen dose (N225 kg/ha^−1^) with a 60% + 40% ratio at jointing stage increased the grain yield and nitrogen utilization rate.

## 1. Introduction

In the world, China is the biggest wheat producing state, however, wheat is the third prominent crop in China after maize and rice. Significant development has been attained in the overall production of wheat and average crop yield per hectare since the establishment of China [1,2]. Nitrogen fertilization is one of the crucial factors that improve the yield and quality of wheat. In recent years, agricultural practices have been motivated to get the most out of yields by the balanced use of N fertilization [3,4] In general, fields are often practiced with a huge amount of nitrogen fertilization, but mostly plants use 5% to 50% of this highly N applied fertilizer in the field. When nitrogenous fertilizers are applied to agricultural fields, few portions are absorbed directly by the plants or others are converted into several other forms through oxidation [5,6]. Farmers in various regions of the world are likely to apply N in additional quantity to achieve higher yields [7]. Extreme N application diminished grain yield and augmented nitrogen loss in a soil ecosystem [8]. Hence, this excess application may cause a nutritional disorder in wheat and limits the crop yield [9]. However, an insufficient amount of nitrogen leads to smaller leaves, poorer chlorophyll content, and reduced biomass production, and consequently diminished grain yield and quality [10]. Thus, nitrogen fertilizers should be used at the right measure and according to the prerequisites of the crops [11]. Consequently, there is a dire need to approach a suitable strategy that ensures the higher grain yield and a drop in ecological environment pollution [12].

Nitrogen is the one of the key components of protein, chlorophyll, and amino acids and have a major involvement in wheat productivity [13]. Wheat grain yield can be augmented by higher biomass and the enhanced harvest index of the wheat [14]. However, the wheat harvest index is now potentially in a plateau and extra improvement in yield will demand an increase in profitable biomass [15,16] Enormous biomass buildup in a photosynthetic area, ultimately leading to an increase in the grain yield potential [17,18]. Grain yield has generally been positively and significantly correlated with total dry biomass production and nutrient uptake in crops. In the meantime, dry matter development and nutrient accumulation differ with growing stages of crops [19]. Nitrogen use efficiency is usually distinct as the dry matter yield per unit of accessible nitrogen [20]. Leaf photosynthetic capacity is a vital character quantifying the crop yield [21,22], which cannot signify the photosynthetic features throughout all the grain-filling stages, particularly at the later stages [23,24]. It has been also reported that the optimum nitrogen rates and timing are important for improving the crop yield and the NUE [25]. A previous study reported that the grain yield increased linearly with the N rates up to 200 kg ha^−1^ with a split application of 50% at sowing time and 50% at tillering stage by increasing the nitrogen uptake and efficiency [26].

Few studies have exposed a positive correlation between nitrogen content and the chlorophyll content of leaves [27]. Therefore, enumerating the chlorophyll content may deliver an indirect determination of the nitrogen status [28]. The interest in exploiting wheat grain yields has stimulated growers to accept demanding management practices. It has been previously reported that high N application increased the vegetative growth and canopy structure of crops, which hinder the light interception resulting in lodging and yield loss [29,30]. On the other hand, the overuse of N also deteriorates the soil properties through nitrogen leaching and volatilization, which effect the nutrient uptake and balance of crops [31]. Therefore, bearing in mind the nitrogen effects on crops, the optimization of nitrogen application at various stages of winter wheat could enhance the nitrogen use efficiency and grain yield. Although several studies have attempted to expose the impact of diverse nitrogen doses and ratios aside from the timing on various developmental stages of multiple crops solely, none of them extensively investigated the all-aforementioned factors together in a single study on winter wheat. Therefore, this study was designed to investigate the impacts of various nitrogen doses, ratios, and timings at various developmental stages of winter wheat. The objectives of this study were as follows: (i) to determine the impact of different nitrogen doses on the LAI (leaf area index), photosynthetic trait, nitrogen uptake, and NUE (nitrogen use efficiency) of winter wheat; (ii) to measure the effects of nitrogen on growth, total biomass production, and grain yield of winter wheat under temperate conditions; and (iii) to assess the association of wheat yield with photosynthetic rate aside from other biomass production traits of winter wheat.

## 2. Material and Methods

### 2.1. Experimental Location and Detail

Study trials were recognized throughout 2017–2018 and 2018–2019 at the Taigu investigational research station of Shanxi Agriculture University, (37°25′ N, 112°33′ E), Province of Shanxi, China. The region of the research area has a moderate central monsoon weather, a mean yearly heat of 13 °C or 12 °C, a mean yearly rainfall of 440 mm and 605 mm, and a probable evapotranspiration of 1840.3 mm and 1872.4 mm, and a sunlight period of 2671 h and 2696 h at the Taigu side, respectively.

The soil characteristics at the research site comprises a clay loam texture and the soil nutrients status is presented in Table 1. The weather data of both growing seasons (2017–2018 and 2018–2019) is displayed in Figure 1.

### 2.2. Soil Sampling and Analytical Methods

The composite soil sample was collected from a 0–20 cm depth. In the laboratory, soil samples were air dried and then passed through a 2-mm sieve size. Later, some of the soil’s fundamental chemical-physical characteristics were measured. Therefore, overall, 30 samples were collected from different sampling points. Forest litter, grass, dead plants, and any other things on the soil surface were removed before sampling, and old manure, moist patches, sites near trees, and compost pits were avoided during sample collection. 

The texture of the soil was determined by the pipette method; soil pH was measured at a soil:water extract ratio (1:5); soil organic matter was measured by the potassium dichromate (K_2_Cr_2_O_7_) wet oxidation method [32]. Total N was measured using the Kjeldhal digestion method. Total P was measured after perchloric and sulfuric acid digestion method [33]. The total K was measured following the earlier published method [34].

### 2.3. Treatments Detail

The experiment was organized in a split plot design by three repeats. The main plots were divided into a total of five nitrogen (N) doses including control (no fertilizer applied), N75, N150, N225, and N300 kg N ha^−1^ and subplots were distributed into the two groups of nitrogen ratios. First, one group was 5:5 (50% + 50%) and the second group was 6:4 (60% + 40%). Different nitrogen doses were applied at various development stages of winter wheat at 0 scale (sowing time), 3 scale (jointing time), 6 scale (flowering time), and 7 scale (grain filling), according to the Zadoks scale in split doses. Two methods were adopted, one method was 50% at sowing time + 50% at jointing, 50% at sowing time + 50% at flowering stage, 50% at sowing time + 50% at grain filling stage and the second method was 60% at sowing time + 40% at jointing stage, 60%, at sowing time + 40% at flowering stage, and 60% at sowing time + 40% at grain filling stage, respectively. The experimental plots were 12 m^2^ (3 m × 4 m) and each treatment had three replications. The nitrogen source was urea 46.4% for the experimental field and applied as a topdressing before sowing and all of the aforementioned growth stages of winter wheat. The phosphorus was applied as triple super phosphate (16%) at 120 kg ha^−1^ and potassium was supplied as the potassium chloride (45% at 60 kg ha^−1^) in the sowing period. The winter wheat variety Jintai 182 was selected due to high acclimatization of the Shanxi Province weather conditions and the sowing rate was 95 kg ha^−1^. Winter wheat was planted on 31 September 2017 and 1 October 2018 and plants were harvested on 15 June 2018 and 21 June 2019, respectively. Data collected from field within the intervals of 20 to 25 days in the months of March, April, and May. All other agronomic practices such as weed controller, irrigation, infection, and insecticide application were managed equally and appropriately on the basis of the wheat growing period and demand using predictable follows of Shanxi Province.

### 2.4. Measurements

#### 2.4.1. Growth and Biomass Characteristics

A winter wheat plant sample was taken at the location where the photosynthetic rate was acquired. A total of 20 cm long winter wheat samples were chosen from all treatments and proximately reserved to the laboratory for dimensions. Afterward, the above ground biomass, leaves, stems, and spikes were measured. An electronic balance was used to determine the balance the fresh weight (g) of winter wheat and converted to the winter wheat each unit per area (N kg ha^−1^). The winter wheat plants were placed in a paper bag, and kept in the oven at a temperature elevated to 105 °C on behalf of thirty min, then back at 80 °C for 24 h to constant weight. The winter wheat plants were placed in a paper bag and kept in an oven at temperature of 105 °C for 30 min and then temperature was set at 80 °C and the dry weight was recorded when the sample weight reached a constant weight. After computing the weight (g) of the dried samples, we converted them into the aboveground biomass, leaves, stems, and spikes dry weight per area (kg m^−2^) using the standard dry weight method [35].

#### 2.4.2. Plant Height (cm)

The height of three winter wheat plants was recorded from downside to the top and the average plant height was calculated.

#### 2.4.3. Leaf Area Index (LAI)

The LAI of wheat plants was determined from jointing to the grain filling stage in two years. Briefly, 15 cm of the wheat plants from the central rows of all plots were destructively collected. To regulate the leaf area of the wheat plants, we distinguished the tall leaf space with a ruler; previously, the leaf area was proposed by increasing the leaf length and the width with a persistence of 0.75 [36], and the proportion of the winter wheat leaf zone to ground zone.

#### 2.4.4. Nitrogen Content Determination

At all-growth stages, the dried wheat plant samples of 20 cm successive plants of winter wheat were ground, subsequently determining the dry matter from each treatment and the nitrogen content of the aboveground biomass, spike grain, stem, and leaf were resolved by using the Kjeldahl method process (Power 1967) after processing for about 0.25 g taster, digestion in sulfuric acid (H_2_SO_4_), and hydrogen peroxide (H_2_O_2)_ [37]. The nitrogen content in diverse winter wheat plant organs were determined by multiplying the total dry matter of each plant organ with the nitrogen content and calculated in g kg ha^−1^. The nitrogen uptake and nitrogen use efficiency can be calculated using the formula below.
$$N\;\mathrm{uptake}\;\left({g\;\mathrm{kg}\;\mathrm{ha}}^{-1}\right)=\frac{\mathrm{Nitrogen}\;\mathrm{content}}{\mathrm{Leaf}\;\mathrm{dry}\;\mathrm{weight}}$$
$$\mathrm{NUE}\;\left(g\right)=\frac{\mathrm{Nitrogen}\;\mathrm{content}}{N\;\mathrm{uptake}}$$

#### 2.4.5. Photosynthesis Measurement

As shown by [38], the photosynthesis rate (Pn) of wheat plants under different growth stages was determined by using the Li-6400 suitable photosynthesis technique (LI-COR Inc. Lincoln, NE USA). From all plots, two winter wheat flag leaves were designated to regulate the photosynthesis rate. The measurements were taken from 10:00 to 11:10 on a strong sunny day below a CO_2_ concentration of 400-mol mol^−1^.

#### 2.4.6. Chlorophyll Measurement

To measure the chlorophyll content, the middle part of the leaves was separated and scratched into minor quantities of 2 mm in thickness. We exactly weighed 0.08 g in a 25 mL volumetric bottle, with an 80% acetone capacity, and placed in a dark environment for 24 h. Chlorophyll a and b were analyzed using an ultraviolet Shimadzu UV-1800 UV (Japan) spectrophotometer at the wavelengths of 665 and 649 nm, respectively [39,40]. 

#### 2.4.7. Yield and Growth Components

Grain yield, grain spike^−1^, spike plot^−1^, and thousand grain weight were determined by harvesting the wheat crop in the central rows from separate treatments. Winter wheat was collected by cutting with a sickle. The spikes were separated from straw grass, recollected in isolated paper bags, oven dehydrated for 24 h at 78 °C, and all samples were threshed manually with a threshing machine. Grains spike^−1^ for all of casually designated plants was computed at the crop harvest and the average was calculated. The thousand grain weight from all plots were selected casuthe ally and weighed to calculate the thousand seed (g). The grain collected from all plots was composed based on the grain yield per plot, and the grain yield ha^−1^ was considered in kilograms. The yield (Gy14% moisture content) and dry substance weight were as described previously [41].
$$\mathrm{Grain}\;\mathrm{yield}\;\left({\mathrm{kg}\;\mathrm{ha}}^{-1}\right)=\frac{{\mathrm{Spike}\;\mathrm{number}\;\mathrm{plot}}^{-1}\times {\;\mathrm{grain}\;\mathrm{number}\;\mathrm{spike}}^{-1}\;}{1000\;\mathrm{grain}\;\mathrm{weight}}$$

### 2.5. Statistical Analysis

The presented data in this research are the mean of three replicates. All data were examined by ANOVA using a randomized complete block design. The significant difference of all foundations was resolved through the F-test. The DMRT significant difference was also measured as a post hoc mean separation (*p* < 0.05) through SAS 9.3. All the treatments were associated based on the significant difference with the least significance difference (LSD *p* < 0.05). All statistical analysis were conducted using SPSS version 20.0 and SAS version 9.3.

## 3. Results

### 3.1. Impact of Nitrogen Application on the Growth Parameters of Winter Wheat

The plant height of winter wheat under all nitrogen fertilizer treatments was significantly higher than that of the control when nitrogen was applied at the jointing stage, under a ratio 60% + 40% (Figure 2). Relative to 150 and 300, under 225 kg N ha^−1^, the plant height was enhanced by 36.50 and 30.63%, and 27.0.3% and 24.94% at the growth stages (the flowering and grain filling stage), respectively. While 150 kg ha^−1^ produced an increment of 10.54% and 11.21%, 13.42 and 14.32%, and 14.10% and 15.07% over 75 and 300 kg N ha^−1^ at (the flowering and grain filling stage), individually. Conversely, the 300 kg ha^−1^ treatment was significantly lower (8%) than for the 225 kg ha^−1^ treatment at the filling stage. Relative to the control, a 7% reduction in plant was observed under 75 kg ha^−1^.

The AGB (aboveground dry biomass) of winter wheat improved gradually throughout the initial jointing growing stage compared to the flowering growth stage, and extended the highest argument at the grain filling stage (Figure 3). When nitrogen was added at the jointing stage, AGB under 225 and 300 kg N ha^−1^ was higher than that of 150 kg N ha^−1^ by 22.62 and 66.90%, and 44.24 and 49.35%, separately, postponing the application of N until the grain filling stage caused a higher AGB under 150 kg N ha^−1^ and 75 kg N ha^−1^ over the control by 18.24% and 20.32%, respectively. However, at a 75 kg ha^−1^ nitrogen dose, the aboveground dry biomass was significantly increased by 27% compared to the control treatment at the flowering stage relative to the jointing stage.

The LAI of winter wheat under all N treatments was significantly greater compared to the control treatment. Among all treatments, the 225 kg ha^−1^ treatment was the most significant at the flowering stage while supply N at the jointing phase under a ratio of 60% + 40% (Figure 4). However, an increment of 80.59% and 71.68%, 58.22% and 54.65%, and 58.98% and 53.40% was observed under 225 kg N ha^−1^ over 75 kg N ha^−1^ and 150 kg N ha^−1^ at the flowering and grain filling stage, respectively.

### 3.2. Effect of Nitrogen on Photosynthesis Rate and Chlorophyll Contents

Compared to the control, the application of N treatments significantly increased the photosynthetic rate of the winter wheat (Figure 5). First, Pn (photosynthetic rate) improved up to the flowering stage, and declined again with the grain filling stage. The 225 and 300 kg N ha^−1^ resulted in the enhancement in the photosynthetic rate by 24.33% and 29.21%, and 22.64% and 25.13% over the 150 kg ha^−1^ treatment, respectively. Moreover, the Pn was enhanced by 11.6% under 150 kg ha^−1^ in relation to 225 kg ha^−1^ at the flowering and grain filling stage, cumulatively.

The chlorophyll content under N fertilizer treatments was also remarkably greater when compared to the control treatment, as depicted in Figure 6. In comparison to the 150 kg N ha^−1^, the chlorophyll content under 225 kg N ha^−1^ and 300 kg N ha^−1^ showed an increasing trend of 57.5% and 105.82%, and 23.57% and 95.92% at the jointing and flowering stages, respectively. The leaf chlorophyll content under 150 at the flowering and grain filling stages was increased by 85.89% and 109.22%, and 75.21, and 80.78%, respectively, compared to the control. The total chlorophyll reduced slowly with the growth development, and showed that the application rate of 150 kg N ha^−1^ was better than that of 75 kg N ha^−1^. Collectively, the application of N fertilizer pointedly improved at the jointing stage in both years, and the efficiency of 225 kg N ha^−1^ was greater than that of all of the other treatments.

### 3.3. Interactive Effect of Nitrogen on Leaf Nitrogen Content, Nitrogen Uptake, and NUE of Winter Wheat

A substantial increment was shown in the leaf N content with the rise in the nitrogen rate (Figure 7). The nitrogen content in the leaves were 43.5% and 47.7% at the jointing stage, 63.5% and 57.3% at the grain filing stage under 60% + 40% ratio, and 37% and 43.5% at the flowering stage under 50% + 50% in the treatment of 225 kg N ha^−1^ in 2018 and 2019, respectively. The maximum results were observed at the 60% + 40% ratio under the 225 kg N ha^−1^ treatment at the jointing stage. However, at the flowering stage, the results ranged from 28.70 to 25.63 under the same treatment, but at the ratio of 50% + 50% where nitrogen was applied at the time of the jointing stage. At the filling stage, the results ranged from 26.34 to 28.34 under the same treatment and 60% + 40% ratio, when nitrogen was applied at the flowering stage in both years compared to 150 and 300 kg N kg ha^−1^, individually (Figure 4).

Nitrogen uptake was also substantially amplified as the nitrogen rates were enhanced (Table 2 and Table 3). Moreover, the results showed that the maximum N uptake (285.06 and 434.76) was noted under the ratio of 60% + 40% compared to the 50% + 50% ratio in 225 kg N ha^−1^ followed by 300 kg N ha^−1^, where the minimum (107.78 and 180.78) N uptake was noted when nitrogen was applied at the time of the jointing and flowering stage compared to the grain filling stage in both years, respectively (Table 3). However, the significant increase in the NUE of winter wheat was recorded when the treatment levels were 75, 150, 225, and 300 kg N ha^−1^ under the ratios of 60% + 40% and 50% + 50% compared to the control. The maximum NUE (0.15 and 0.11) was noted where the 150 kg N ha^−1^ and 50% + 50% ratio was practiced and the minimum N use efficiency (0.09 and 0.10) was observed where no N fertilizer was supplied at the jointing stage compared to the flowering and grain filling stage (Table 2).

### 3.4. Impact of Nitrogen on Yield and Yield Components of Winter Wheat

The records pertaining to the grain spike^−1^ of wheat was impacted by altered levels of nitrogen (Table 4). The results showed that the nitrogen dose at 225 kg N ha^−1^ under the ratio of 60% + 40% when the N was applied at the time of the jointing stage produced significantly maximum grains per spike of 46 and 33.6 followed by 300 kg N ha^−1^ in 2018 and 2019, respectively, although the minimum grains per spike (26 and 19.7) were observed in the control plots in 2018 and 2019, respectively. Overall, the highest results were observed at the 60% + 40% ratio compared with 50% + 50% under all treatments, as shown in Table 4. The results showed that 225 kg N ha^−1^ under the ratio of 60% + 40% produced the maximum seed index of 43.07 and 42.33 g, followed by 150 kg N ha^−1^ (38.53 and 39.57 g) in 2018 and 2019, respectively, when the N was supplied at the jointing stage. However, the minimum seed index (34 and 34.5 g) was observed in the control plots in 2018 and 2019, respectively.

Moreover, the results showed that 225 kg N ha^−1^ under the ratio of 60% + 40% produced the maximum grain yield (7316 and 6228 kg ha^−1^) followed by 300 kg N ha^−1^ (60100 and 6180 kg N ha^1^) in 2018 and 2019, respectively (Table 4). Compared to 225 kg N ha^−1^, 75 and 150 kg N ha^−1^ showed a reduction (5106.07 to 6029.03 and 5763.03 to 6236.56 kg N ha^−1^) in the grain yield. On the other hand, the lowest grain yield (4561.00 to 5406.33 kg ha^−1^) was perceived in the control plots. Taken altogether, the higher results were observed at the 60% + 40% ratio when the nitrogen was applied at 225 kg ha^−1^. Hence, this study revealed that the crop sown with the optimum nitrogen rate resulted in a positive impact on all of the growth and yield contributing traits.

### 3.5. Relationship between the Studied Parameters

After the Pearson’s association examination, a strongly positive relationship was revealed between the grain yield and photosynthetic rate, nitrogen content, chlorophyll, nitrogen uptake, spike plant^−1^, and seed index (Figure 8), although a destructive relationship was demonstrated by the yield and NUE. Furthermore, only a minor positive relationship was revealed between the yield and dry biomass, plant height, and grains spike^−1^.

## 4. Discussion

### 4.1. Interactive Effect of Nitrogen on Growth Parameters of Winter Wheat

Nitrogen can significantly affect the plant growth and development as a key macronutrient. The findings of our study revealed that the plant height of winter wheat was lowest under the control treatment when compared to different nitrogen treatments and ratios. This is an agreement with previous findings reported by Tanka et al. where in the first winter season, the plant height improved and reached a high of more than 120 cm at the milk ripping period, and in the second winter season, the plant height extended over 100 cm at flowering and 120 cm at the grain filling period [42]. Furthermore, the differences in the two growing season were mainly attributed to the better development of winter wheat in the 2017–2018 growing season, which resulted in the right time of top-dressing application. Our findings were in line with those by Yingkui et al., who noted that the application of different ratios of fertilizer ranging from 70:10:20:0%, 50:10:20:20%, and 30:10:30:30% had a significant effect on winter wheat [43].

Aboveground dry biomass and the LAI of winter wheat crops showed significant differences (*p* < 0.05) that were correlated with the jointing to the grain filling time under different nitrogen doses and nitrogen ratios (Figure 2 and Figure 3). The application of nitrogen fertilizer significantly improved the dry biomass and LAI of the winter wheat under all growing stages. Our results revealed that the 225 kg N ha^−1^ nitrogen application rate at the 60% + 40% ratio improved the AGB by 27.45% at the grain filling stage. Earlier studies have shown that an appropriate LAI was advocated to be 5 to 7 throughout the flowering stage [44,45]. Certain studies have shown that when the LAI is around 3, the radiation capture of PAR attitudes is 90%; any further increase in LAI would be useless [46] since the extreme LAI will decrease the light concentration and/or since the light quality changes at the inadequate of the canopy, where tiller shoots and new tillers are situated [47,48]. However, in this investigation, the LAI was improved under 225 kg N ha^−1^ at the flowering growing stage under the ratio of 60% + 40% when nitrogen was added at the time of the jointing period. An appropriate LAI 5 to 7 was achieved below the appropriate nitrogen supply conditions of 180 kg N ha^−1^ at the jointing to heading stage, thus the optimum leaf biomass portion of AGB might be a significant applicant to determine the yield. In accordance with our results, another study reported that the leaf biomass segment of the aboveground biomass improved with the nitrogen supply in a positive range and that the ratio was slightly greater at the heading development [44]. Collectively, we observed a significant effect of N fertilizer rates and N ratios on the dry biomass and LAI. Specific studies showed that the growth investigation verified significant relations between the growth rates and grain yield at both the separate plant and plant position level. This is in agreement with our outcomes, showing a significant correlation between the growth rate and grain yield [42,49].

### 4.2. Impact of Nitrogen on Photosynthetic Rate and Chlorophyll Content

The Pn is a vital characteristic of photosynthesis and improving the photosynthetic rate could ultimately enhance the grain yield of the winter wheat plants. The Pn was impacted by nitrogen fertilizer and appropriate consumption of the N was significantly enhanced in winter wheat [50]. In the present study, N timings, nitrogen rates, and ratios substantially enhanced the Pn at the flowering and the grain filling stages in both of the studied years. Moreover, a higher Pn was noticed under a ratio of 60% + 40% (treatment of 225 kg N ha^−1^) at the jointing, flowering, and filling periods when the nitrogen was applied at the jointing stage (Figure 4). At all growth periods, the change in the average Pn below the two topdressing fertilizer managements showed that the flowering stage was greater than the jointing and filling stages. These outcomes confirmed that by N splitting application and delaying N quantity, a moderately higher Pn can be preserved at the later growth periods of winter wheat and that the flowering stage is the best stage fertilizer application as topdressing. Prominently, the application timing impact on the Pn of winter wheat was apparent, and a higher Pn rate was measured under the flowering and grain filling stage, which was related to the greater transmittance under early growing stages that assisted the plants in increasing their leaf area and accumulating additional sunshine. This might be due to promising climate conditions and greater chlorophyll content, leaf area index, and net photosynthetic rate [51,52]. The increment in the photosynthetic rate was noticed only because the collection of the optimum application timing noticeably enhanced the LAI and nitrogen uptake by the winter wheat crop. However, a reduction in the Pn of winter wheat in the control was credited to decrease the leaf area and nitrogen uptake. Consequences from the present research have effects for both the agronomic and conservational aspects of N fertilization management [53].

Chlorophyll is an important pigment of photosynthesis and a virtuous indicator of the leaf functions under the damaging effects of different ecological agents [49], and it is also a vital parameter for witnessing the uptake of nitrogen in winter wheat [42]. Our outcomes exhibited that, relative to the untreated plants, the total Chl contents were initially augmented with an increment in the nitrogen rate and then declined in both years (Figure 6). Prost and Bao et al. reported that the total Chl and carotenoid contents of the winter wheat plants were greater than CK at the jointing to grain-filling periods [54,55]. Comparable results have been described by a number of studies in different types of crops [56]. Nitrogen has a number of functions such as an increase in the photosynthetic rate, an improvement in the tissue strength, and a decline in the plant transpiration rate. Our results demonstrated that the total Chl was highest under 225 kg N ha^−1^ at the 60% + 40% ratio at the jointing stage, when nitrogen was added at the time of the jointing period. Similarly, Du et al. described that the chlorophyll content of the winter wheat leaves significantly proliferated at the jointing and booting growth stages compared to that in the control (*p* < 0.05) [51]. A previous study [57] revealed that the leaf chlorophyll content under jointing and flag leaf at various periods was enhanced by 3.3 to 6.8 and 3.0 to 23.4%, respectively, compared to that under the early flowering stage when applying nitrogen at the jointing stage. This outcome shows that the topdressing nitrogen application is in close competition with the nitrogen supplies of both the N timing and treatments, and the ratios of winter wheat finally add to the dry matter accumulation, subsequently to a greater N supply in the leaves, finally leading to significantly greater grain yields.

### 4.3. Effect of Nitrogen on Leaf Nitrogen Content, Nitrogen Uptake, and NUE

The nitrogen concentration in leaves significantly enhanced with an increase in the nitrogen application (Figure 6). Among all treatments, the maximum results were observed at the ratio of 60% + 40% and 225 kg N ha^−1^ treatment at the jointing stage. Our results are inconsistent with the previous investigation of Tanka et al., who described that the highest total nitrogen content in leaves ranged from 30 to 38 (g kg^−1^) was observed at the jointing to flowering stage of wheat crop [58]. Our findings revealed that the application of nitrogen fertilizers at the sowing time was favorable for the proliferation of the leaf nitrogen content. Our results were comparable to the effort of Jones et al. and Feng et al., who stated that nitrogen fertilizer had a significant influence on the early wheat plant nitrogen content at the main development stages [27]. High doses of nitrogen fertilizer repressed the root development and root activity, and therefore reduced the uptake and absorption of nitrogen at the growing stage in our present study. In addition, our results were also in accordance with the earlier trial conducted by Sower et al., who reported that a certain quantity of nitrogen fertilizer was favorable to the root growth and thus better nitrogen uptake, particularly in soil with less available nitrogen [59,60,61,62,63,64]. It was further noticed that the nitrogen uptake of winter was significantly impacted by different application ratios and nitrogen rates and their timings. The maximum nitrogen uptake was found at 225 kg N ha^−1^ under the ratio of 60% + 40%; similar results were also observed in previous studies (Gupta 2014, Jiang 2018, [60]). According to our outcomes, the NUE declined with an increasing nitrogen rate (Table 2). Similarly, Sower et al. also identified that, in general, the NUE reduced with an improvement in the nitrogen fertilizer rate, but increased the yield and N loss [65,66,67,68,69].

### 4.4. Impact of Nitrogen Application on Yield and Yield Components

In our study, the grain yield of winter wheat exposed a significant response (*p* < 0.05) to diverse the timings, treatments, and ratios of N fertilization. Among the treatments, the 225 kg N ha^−1^ contributed a significant enhancement to the grain yield and yield components under the r60% + 40% ratio when nitrogen was applied at the jointing stage compared to the flowering and grain filling stages (Table 3). These results are in agreement with those of [70,71], who showed that the grain yield characteristics were impacted by the N rates and the ratios of pre-sowing to topdressing fertilizer, and that crop yield was generally more dependent on the fertilizer rates and fertilization timing. Our results confirm that some previous studies have highlighted that nitrogen nutrition has a significant impact on plants and the increased doses of nitrogen had a noteworthy influence on winter wheat plants up to 200 and 225 kg N ha^−1^ [50,72]. Collectively, our findings indicate that nitrogen has an essential role in the growth and development of grain, which were also previously confirmed by other studies [3,21]. Our results advocate the importance of balanced nitrogen nutrition for winter wheat crops. Several other studies have also documented the positive response of various crops to N nutrition in terms of growth, yield, and nitrogen fertilization (e.g., wheat, rice, maize [18,64,65,66,67,68,69,70,71,72,73]). Sustainable and optimizing crop production depends on the continuous redevelopment of soil fertility over a balance between the nitrogen supply and demand of cropping systems. Based on the overall outcomes of this study, the following are some recommendations such as the application of 225 kg N ha^−1^ is encouraged when the aim is to improve the photosynthesis and yield. However, to improve the nitrogen use efficiency, it can be recommended to use 75 kg N ha^−1^ to 300 kg N ha^−1^ and 225 kg N ha^−1^ under experimental conditions.

## 5. Conclusions

The outcomes of our study concluded that there was a strong positive relationship between the chlorophyll, photosynthetic rate, N content, N uptake, grain yield, spike plant^−1^, and seed index under different nitrogen doses and ratios. Moreover, the 225 kg ha^−1^ dose and ratio of the split application of nitrogen (60% + 40%) effectively promoted the biomass traits, photosynthetic characteristics, nitrogen uptake, and yield components of the winter wheat at the jointing stage. Our study exposed that the Pn was enhanced up to the flowering stage, and decreased with the grain filling stage under different nitrogen doses and ratios. However, the results showed that the N content in leaves were enhanced by 43.5% and 47.7% at the jointing stage, followed by 225 kg N ha^−1^ at 63.5% and 57.3% at the grain filing stage under the 60% + 40% ratio in 2018 and 2019, respectively. Furthermore, the N uptake and NUE were prompted at a nitrogen dose of 225 kg N ha^−1^ and ratio of 60% + 40% when compared to the control treatment at the jointing and grain filling stages. Therefore, our findings optimize and recommend the 225 kg N ha^−1^ dose of nitrogen and the ratio of 60% + 40% at the jointing stage for the maximum Pn, dry biomass accumulation, NUE, and an ultimately higher grain yield in winter wheat.

## Figures and Tables

**Figure 1 life-12-01478-f001:**
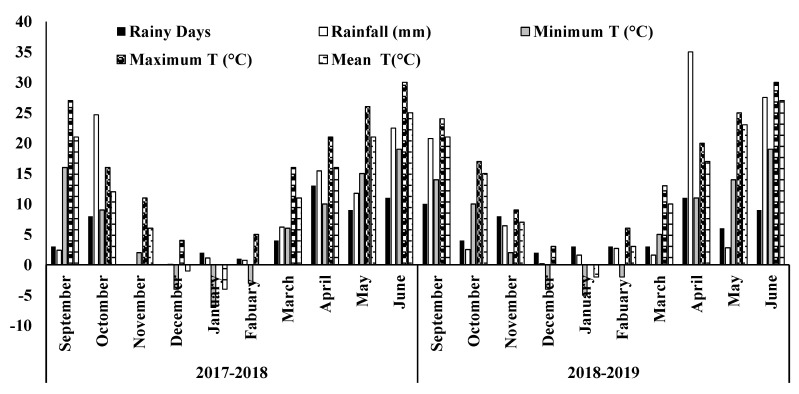
Monthly rainy day, rainfall, minimum temperature, maximum temperature, and mean temperature of 2017–2018 to 2018–2019 at the experimental site of Shanxi agricultural university.

**Figure 2 life-12-01478-f002:**
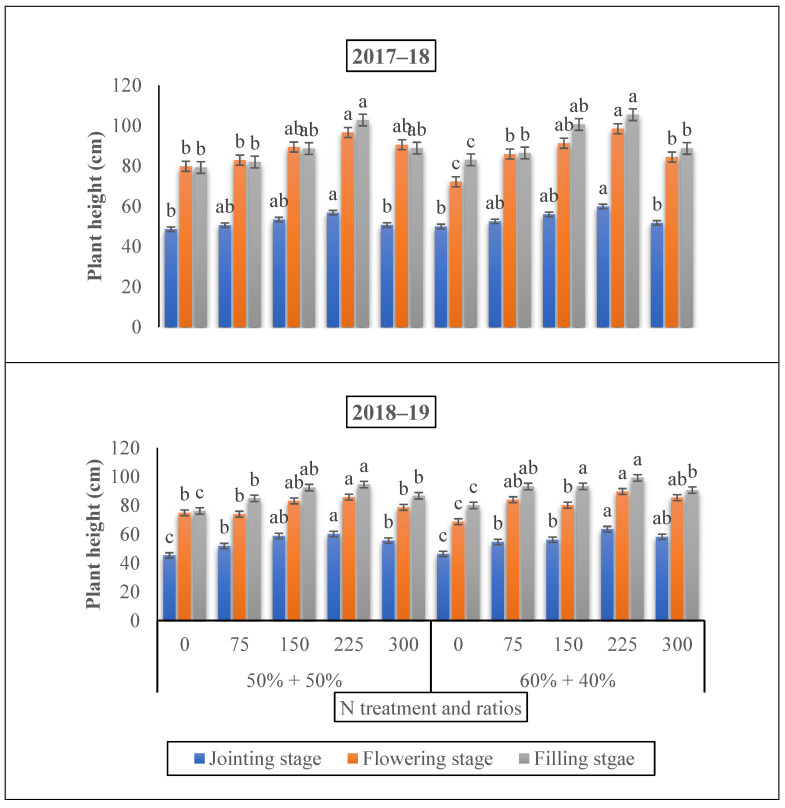
The effect of nitrogen fertilizer application on the plant height (cm) of winter wheat at various developmental stages. Mean values in a separate column followed by similar letters were not significantly different at *p* < 0.05. The values are the means ± SE (standard error).

**Figure 3 life-12-01478-f003:**
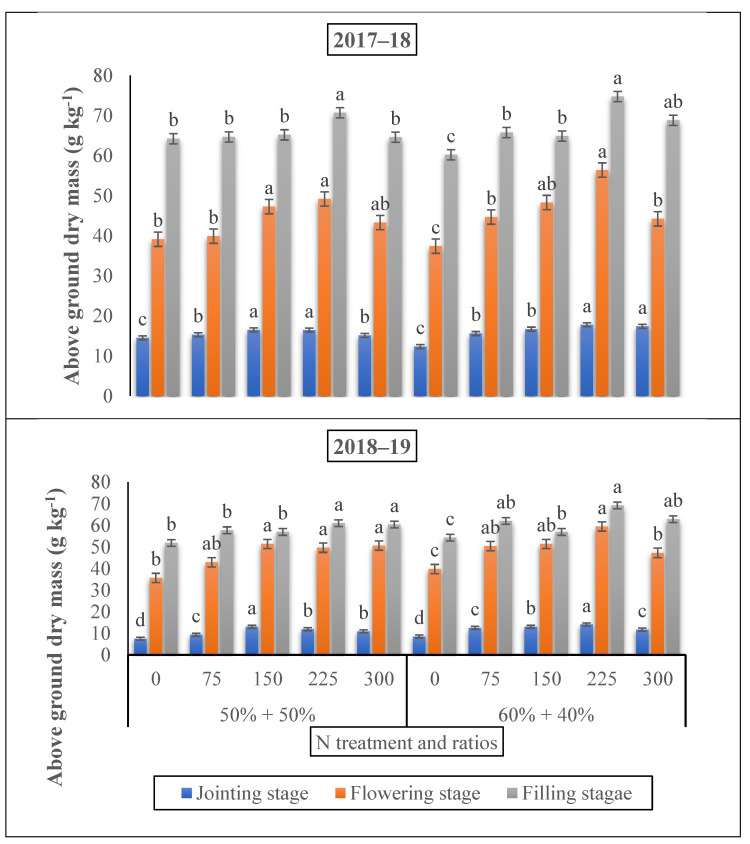
The effect of nitrogen fertilizer application on the aboveground biomass (g kg^−1^) of winter wheat at various developmental stages. Mean values in a separate column followed by similar letters were not significantly different at *p* < 0.05. The values are the means ± SE (standard error).

**Figure 4 life-12-01478-f004:**
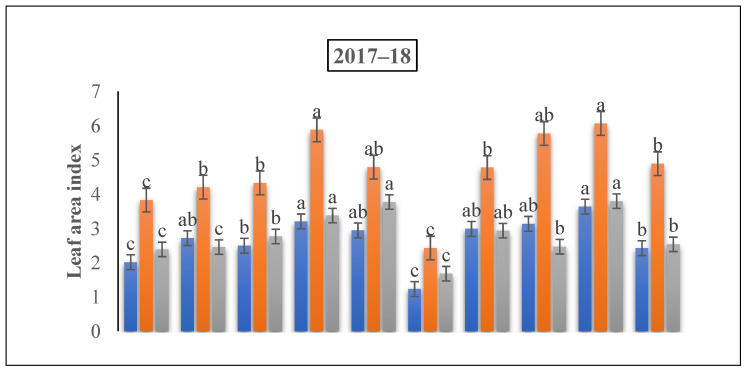
The effect of nitrogen fertilizer application on the leaf area index of winter wheat at various developmental stages. Mean values in a separate column followed by similar letters were not significantly different at *p* < 0.05. The values are the means ± SE (standard error).

**Figure 5 life-12-01478-f005:**
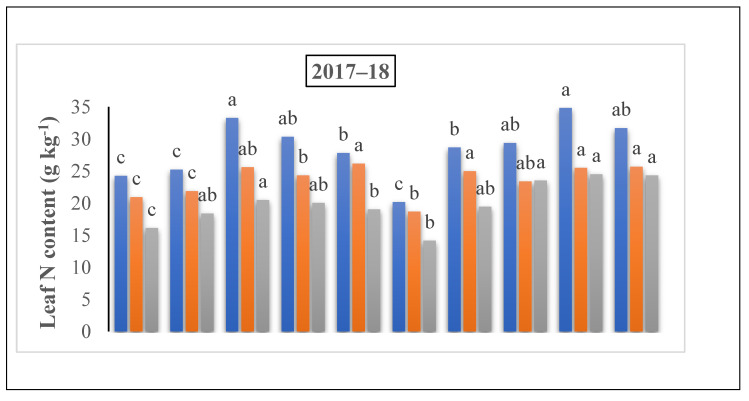
The effect of nitrogen fertilizer application on the N content in leaves (g kg^−1^) of winter wheat at various developmental stages. Mean values in a separate column followed by similar letters were not significantly different at *p* < 0.05. The values are the means ± SE (standard error).

**Figure 6 life-12-01478-f006:**
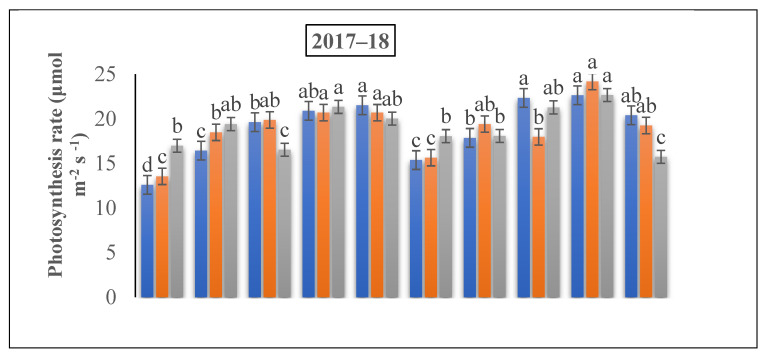
The effect of nitrogen fertilizer application on the photosynthetic rate of winter wheat at various developmental stages. Mean values in a separate column followed by similar letters were not significantly different at *p* < 0.05. The values are the means ± SE (standard error).

**Figure 7 life-12-01478-f007:**
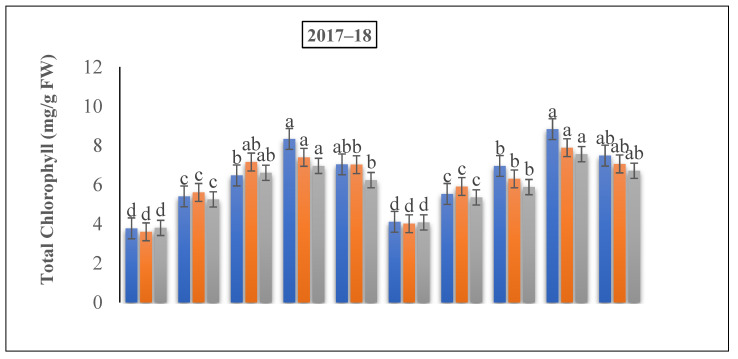
The effect of nitrogen fertilizer application on the total chlorophyll content of winter wheat at various developmental stages. Mean values in a separate column followed by similar letters were not significantly different at *p* < 0.05. The values are the means ± SE (standard error).

**Figure 8 life-12-01478-f008:**
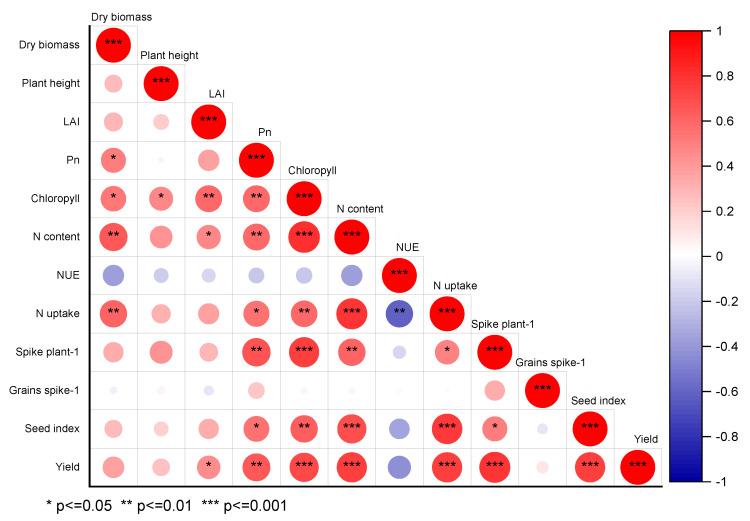
The relationship between the yield and other studied parameters. Red represents a positive correlation, and blue represents a negative correlation (* *p* ≤ 0.05, ** *p* ≤ 0.01, *** *p* ≤ 0.001). The intensity of color represents the significance of a variable. The LAI, Pn, N, and NUE refer to the leaf area index, photosynthesis rate, nitrogen, and nitrogen use efficiency, respectively.

**Table 1 life-12-01478-t001:** The soil properties prior to the experiments at the farming station of the Shanxi agricultural university.

Year	Total N (g kg^−1^)	Total P (g kg^−1^)	Total K (g kg^−1^)	SOM (g kg^−1^)	pH
2017–2018	51.12	19.34	143.26	7.98	7.7
2018–2019	49.09	16.21	134.06	7.56	7.2

**Table 2 life-12-01478-t002:** The effect of the nitrogen fertilizer application on the yield, N uptake, and NUE of winter wheat.

	N Treatment	50% + 50% N Fertilizer Ratio			60% + 40% N Fertilizer Ratio		
		Yield kg	N Uptake	NUE	Yield kg	N Uptake	NUE
**2017–2018**	**0**	4561.0 ± 6.6 d	132.15 ± 2.86 d	0.14 ± 0.00 a	4451.0 ± 6.4 d	107.78 ± 4.82 d	0.09 ± 0.00 c
	**75**	4984.3 ± 8.2 c	141.80 ± 5.00 c	0.13 ± 0.00 a	5106.7 ± 9.9 c	145.59 ± 4.32 c	0.13 ± 0.00 a
	**150**	5370.2 ± 9.8 b	155.01 ± 4.43 c	0.13 ± 0.01 b	5763.0 ± 5.8 c	171.20 ± 4.43 c	0.12 ± 0.01 b
	**225**	6390.8 ± 9.6 a	205.85 ± 1.78 a	0.10 ± 0.00 c	7316.6 ± 7.7 a	285.06 ± 4.70 a	0.09 ± 0.01 c
	**300**	5834.0 ± 5.7 b	187.71 ± 5.14 b	0.12 ± 0.01 a	6010.8 ± 4.7 b	206.16 ± 5.74 b	0.10 ± 0.00 b
**2018–2019**	**0**	5406.3 ± 3.9 d	180.78 ± 3.87 d	0.10 ± 0.00 b	5200.3 ± 2.7 d	102.72 ± 4.80 c	0.06 ± 0.00 c
	**75**	5925.7 ± 8.3 c	203.91 ± 4.65 d	0.10 ± 0.01 b	6029.0 ± 9.6 b	236.05 ± 3.23 b	0.09 ± 0.01 a
	**150**	6133.2 ± 6.3 a	330.53 ± 2.56 b	0.11 ± 0.01 a	6236.5 ± 8.3 a	215.63 ± 2.65 b	0.09 ± 0.02 a
	**225**	6225.1 ± 3.5 a	401.84 ± 2.54 a	0.06 ± 0.01 d	6328.4 ± 4.9 a	311.28 ± 5.54 a	0.07 ± 0.0 b
	**300**	6086.8 ± 8.5 b	295.61 ± 1.77 c	0.07 ± 0.01 d	6190.1 ± 8.8 b	283.92 ± 4.42 b	0.07 ± 0.00 b

Means in separate columns followed by similar letters were not significantly different at *p* < 0.05. The values are the mean ± SE (standard error).

**Table 3 life-12-01478-t003:** The significance of the F-value from the analysis of variance of various parameters of winter wheat under different nitrogen rates and ratios at various developmental stages.

Parameter	N-Rates (N)	Ratios (R)	N × R
Plant height	539.57 ***	39.90 ***	6.77 ***
Plant dry biomass	191.36 ***	22.08 ***	17.46 ***
Leaf area index	58.40 ***	1.51 NS	5.79 ***
Leaf N content	161.38 ***	6.19 ***	5.20 ***
Photosynthetic	101.01 ***	2.95 *	4.80 ***
Total Chl	71.05 ***	5.75 ***	2.10 **
Spike plant^−1^	76.01 ***	0.72 NS	0.60 NS
Grains Spike^−1^	544.35 ***	10.90 ***	5.62 ***
1000 seed weight	267.86 ***	31.77 ***	9.33 ***
Yield (kg ha^−1^)	1656.07 ***	15.99 ***	14.43 ***
N Uptake	54.67 ***	2.06 *	1.81
NUE	45.84 ***	0.83 NS	1.63 **

**Note:** *, **, and *** denote the significance levels at alpha 0.05, 0.01, and 0.001 obtained by morality. Significant difference (HSD) test. ‘NS’ denotes non-significance.

**Table 4 life-12-01478-t004:** The effect of nitrogen fertilizer application on the spike, grains, and 1000 seed weight of winter wheat.

	N Treatment	50% + 50% N Fertilizer Ratio			60% + 40% N Fertilizer Ratio		
		Spike Plant^−1^	Grains Spike^−1^	1000 Seed Weight (g)	Spike Plant^−1^	Grains Spike^−1^	1000 Seed Weight (g)
**2017–2018**	**0**	291.33 ± 5.57 d	26.00 ± 012 d	34.40 ± 0.32 c	288.30 ± 4.66 d	24.00 ± 002 d	32.38 ± 0.30 c
	**75**	310.00 ± 4.53 cd	28.00 ± 0.19 c	34.43 ± 0.46 b	321.67 ± 6.09 c	30.67 ± 1.11 c	40.73 ± 0.95 ab
	**150**	361.33 ± 4.76 c	35.67 ± 0.75 b	35.90 ± 0.37 a	363.33 ± 3.65 c	38.67 ± 0.70 c	36.17 ± 0.17 b
	**225**	502.33 ± 2.98 a	39.33 ± 0.22 a	35.27 ± 0.17 a	584.00 ± 6.23 a	46.00 ± 0.09 a	43.07 ± 0.15 a
	**300**	430.00 ± 6.43 b	39.33 ± 0.58 a	33.47 ± 0.92 b	468.67 ± 5.20 b	40.67 ± 0.12 b	35.30 ± 0.15 b
**2018–2019**	**0**	276.33 ± 3.53 d	19.70 ± 0.58 d	34.63 ± 0.56 d	273.23 ± 3.44 d	17.70 ± 0.54 d	31.50 ± 0.56 d
	**75**	294.33 ± 1.45 cd	21.50 ± 0.58 c	37.27 ± 0.22 c	318.33 ± 3.76 c	25.15 ± 0.55 c	37.50 ± 1.47 c
	**150**	382.00 ± 2.08 b	27.63 ± 0.67 b	39.33 ± 0.37 b	397.00 ± 3.93 ab	28.30 ± 0.33 b	39.57 ± 0.17 b
	**225**	412.67 ± 3.84 a	29.63 ± 0.33 a	40.47 ± 1.12 a	432.67 ± 5.51 a	33.65 ± 0.38 a	42.33 ± 0.23 a
	**300**	341.67 ± 3.23 c	27.40 ± 0.33 b	38.37 ± 0.18 b	367.33 ± 4.48 b	28.30 ± 0.88 b	39.07 ± 0.15 b

Means in separate columns followed by similar letters were not significantly different at *p* < 0.05. The values are the means ± SE (standard error).

## Data Availability

Not Applicable.

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
