# Peer review of "Improving Winter Wheat Photosynthesis, Nitrogen Use Efficiency, and Yield by Optimizing Nitrogen Fertilization"

_life, 2022, doi:10.3390/life12101478_

Round 1

Reviewer 1 Report

Comments and Suggestions for Authors

1. There are some grammatical errors in the manuscript, further check by native English speaker is necessary.

2. The title of the paper is suggested to be revised.

3. The abstract does not report the main findings of the study in a clear manner and need to re-write. For example, general expressions are used which do not provide useful information to the readers. Information that is more specific is required in the abstract. In addition, the author failed to successfully convey to the reader what the author wanted to convey in the abstract. Whats the mean of this sentences such as “a field study was established with five N treatments (0 control, N75, N150, N225 and N300 kg ha-1) prepared as main plots and two N ratios (5:5 and 6:4) in subplots at pre-sowing, jointing, flowering, and the grain filling stages.” and “leaf area was enhanced by 149% at flowering stage under 60% + 40%.

4. Why is the comparison of dry matter at grain filling stage and not at maturity in this study? The dry matter accumulation at maturity determines the yield.

5. The introduction does not clarify the importance of conducting this study and needs to be supplemented with the scientific issues and related advances in this study. For example, what are the current advances in research on N application rate, time and N management, and its effects on wheat yield, N use efficiency and the related mechanisms? Overall, the introduction does not point out the gap of this study seeks to fill and novelty of the study over the existing literature. This point showed be further elaborated.

6. L72-74 Whats the mean of “High nutrient concentration can also damage crops by creating wheat plants more susceptible to lodging, affecting both compensations to the environment through leaching [27] and nitrogen volatilization [28], and economic damages to farmers [29].”? leaching and nitrogen volatization?? There are many unclear descriptions or errors in the manuscript, and the authors are advised to carefully check and revise the entire text word by word.

7. In the Material and methods part, the description of the experimental design is not clear, please further refinement. In addition, how the N fertilizer was applied to the field needs to be added, especially in the later growth stages of wheat, how was the N fertilizer applied into the field in this stages? Whether the timing of nitrogen application was consistent in both years? Please use Zadoks scale to indicate growth stages.

8. L 108 The experiment were organized in a split plot design by three repeats. L115-116 The

9. experimental plots were 12 m2 (3m × 4m), and each treatment had nine replications. What this mean? Why only Jintai 182 was chosen in this study? Is it the main cultivar cultivated in the area? The information needs to be added. It is better to provide the density of wheat after seedling emergence rather than the amount of seeds sown, because the sowing date and germination rate have an impact on wheat emergence. What is the wheat row spacing? Sampling and data collection time should be specific to the wheat growth period. Irrigation has a significant impact on N use and two years of irrigation schedule needs to be added.

10. L136  back at 80°C for 24 hours to constant weight. Can such conditions bring the wheat plant to constant weight? Especially for the later growth period of wheat samples? The authors are advised to be realistic in their presentation of the experimental procedure.

11.  More information should be written about the chlorophyll determination method, at which wavelength the chlorophyll analyzes were performedand this information should be supported by the literature. Which spectrophotometer was used in the study (company, country)?

12. Why is the theoretical yield presented in the study results instead of the actual measured yield? The theoretical yield calculation formula in L180 is incorrect? Please check it.

13. In the Results part, the authors were advised to merge simple results or make them into one table, and to further summarize and condense the findings without over-describing non-critical  experimental results. And the tables in the manuscript are not standardized enough and need to be revised.

14. The discussion section should discuss and analyze the mechanism of the important findings obtained in this study, rather than re-description of the findings. In addition, it is recommended that an important discussion should be carried out for the important findings of the study, while simple results such as plant height and some common sense results do not need to be discussed with too much text. It is recommended to discuss and analyze the mechanism of changes in dry matter production, photosynthetic capacity, grain yield and N use efficiency of wheat by the amount and period of N application, and to clarify the relationship between these indicators and the amount and time of N fertilizer application.

Author Response

  1. There are some grammatical errors in the manuscript, further check by native English speaker is necessary.

Response: Thank you very much for your careful reading and sincere efforts to improve our work. We have checked all the errors throughout the manuscript and modified.

  1. The title of the paper is suggested to be revised.

The title of paper has been revised.

  1. The abstract does not report the main findings of the study in a clear manner and need to re-write. For example, general expressions are used which do not provide useful information to the readers. Information that is more specific is required in the abstract. In addition, the author failed to successfully convey to the reader what the author wanted to convey in the abstract. What’s the mean of this sentences such as “a field study was established with five N treatments (0 control, N75, N150, N225 and N300 kg ha-1) prepared as main plots and two N ratios (5:5 and 6:4) in subplots at pre-sowing, jointing, flowering, and the grain filling stages.” and “leaf area was enhanced by 149% at flowering stage under 60% + 40%.

We have modified the abstract according to the suggestions.

  1. Why is the comparison of dry matter at grain filling stage and not at maturity in this study? The dry matter accumulation at maturity determines the yield.

Thank you for your suggestion. Since our main focus was to investigate the perforamnce of winter wheat at three various stages i.e. jointing, flowering and grain filling stages. That is the reason why we didn’t measure the dry mass at maturity stage for this stage. But, we are focusing it in our another on-going project.

  1. The introduction does not clarify the importance of conducting this study and needs to be supplemented with the scientific issues and related advances in this study. For example, what are the current advances in research on N application rate, time and N management, and its effects on wheat yield, N use efficiency and the related mechanisms? Overall, the introduction does not point out the gap of this study seeks to fill and novelty of the study over the existing literature. This point showed be further elaborated.

It has been updated in the manuscript.

  1. L72-74 What’s the mean of “High nutrient concentration can also damage crops by creating wheat plants more susceptible to lodging, affecting both compensations to the environment through leaching [27] and nitrogen volatilization [28], and economic damages to farmers [29].”? leaching and nitrogen volatization?? There are many unclear descriptions or errors in the manuscript, and the authors are advised to carefully check and revise the entire text word by word.

The following sentence have been modified and all errors resolved throughout the manuscript.

  1. In the Material and methods part, the description of the experimental design is not clear, please further refinement. In addition, how the N fertilizer was applied to the field needs to be added, especially in the later growth stages of wheat, how was the N fertilizer applied into the field in this stages? Whether the timing of nitrogen application was consistent in both years? Please use Zadoks scale to indicate growth stages.

Thank you for your careful reading. We have refined the materials and methods section. Nitrogen was applied through manuall topdressing at various developmental stages throughout the experiment. And the timing of nitrogen application was consistant in both growing seasons.

We have also indicated the growth stages according to the Zadoks scale in the materials and methods section.

  1. L 108 “The experiment were organized in a split plot design by three repeats.” L115-116 “The 9. experimental plots were 12 m(3m × 4m), and each treatment had nine replications.” What this mean? Why only Jintai 182 was chosen in this study? Is it the main cultivar cultivated in the area? The information needs to be added. It is better to provide the density of wheat after seedling emergence rather than the amount of seeds sown, because the sowing date and germination rate have an impact on wheat emergence. What is the wheat row spacing? Sampling and data collection time should be specific to the wheat growth period. Irrigation has a significant impact on N use and two years of irrigation schedule needs to be added.

The experiment was designed in split plot design with three replications. There is mistake in writing for nine replication and we have modified the sentence. The winter wheat Jintai 182 cultivar was selected in this study, because it is local cultivar of Shanxi province and highly adaptive to its environment.

Since our main focus of the study was to improve the nitrogen use effeciency and yield of this cultivar. However, the germination rate of this cultivar was 85-90%, so, measuring of density of wheat after seedling emergence was not focused. And all other practices were adopted according to local witner wheat farming practices like row spacing (wheat sown through machine), irrigation. Sampling and data collection had mentioned in the manuscript.

  1. L136  “back at 80°C for 24 hours to constant weight.” Can such conditions bring the wheat plant to constant weight? Especially for the later growth period of wheat samples? The authors are advised to be realistic in their presentation of the experimental procedure.

Thank you for your careful reading. We have modified the method in the manuscript.

  1. More information should be written about the chlorophyll determination method, at which wavelength the chlorophyll analyzes were performedand this information should be supported by the literature. Which spectrophotometer was used in the study (company, country)?

Chlorophyll a and chlorophyll b were measured by using acetone (80%) and impurities were detached by centrifugation. Chlorophyll a and b were analyzed using an ultraviolet Shimadzu UV-1800 UV (Japan) spectrophotometer at the wavelengths of 665 and 649 nm, respectively. We have added this information in the manuscript.

  1. Why is the theoretical yield presented in the study results instead of the actual measured yield? The theoretical yield calculation formula in L180 is incorrect? Please check it.

Thank you for your valuable suggestion, but we have calculated the theoratical yeild according to our recently published study (Kubar, M.S.; Zhang, Q.; Feng, M.;Wang, C.; Yang,W.; Kubar, K.A.; Riaz, S.; Gul, H.; Samoon, H.A.; Sun, H.; et al. Growth, Yield and Photosynthetic Performance ofWinter Wheat as Affected by Co-Application of

Nitrogen Fertilizer and Organic Manures. Life 2022, 12, 1000.).

  1. In the Results part, the authors were advised to merge simple results or make them into one table, and to further summarize and condense the findings without over-describing non-critical  experimental results. And the tables in the manuscript are not standardized enough and need to be revised.

Results section is modified according to the suggestions and tables are standerized, too.

  1. The discussion section should discuss and analyze the mechanism of the important findings obtained in this study, rather than re-description of the findings. In addition, it is recommended that an important discussion should be carried out for the important findings of the study, while simple results such as plant height and some common sense results do not need to be discussed with too much text. It is recommended to discuss and analyze the mechanism of changes in dry matter production, photosynthetic capacity, grain yield and N use efficiency of wheat by the amount and period of N application, and to clarify the relationship between these indicators and the amount and time of N fertilizer application.

We have modified the discusion section according to the suggestions.

Reviewer 2 Report

Abstract: add an initial phrase of introduction.

Explain the N rates in the abstract; what is the means for 5:5?

Line31: Check it “N225 kg ha-1”. What is N225 ?

The abstract could have more information about the experiment. Explore more details about the site.

Line 38: race crop?

In the Introduction could explore the N rates used commonly. There are few information about N management in wheat. Explore it.  

Line82: what is LAI? NUE?

The first paragraph in the Material and Methods is confused. Please, divide it.

Soil characterization? Soil layer? Nutrient contents? Citation of soil methodologies used in characterization. How many samples?

Explain how N was applied in soil.

Line 108-126: too long. Please, divide it.

Is there lime application? Yes, rate? When? Or Why not?

Line 200: What is AGB?

The results of dry mass and height were not expected in the various developmental stages? Explain the results clearly because it is expected higher development with older plants.

Line 229: what is Pn?

Line 230-231: “Pn improved up to the flowering stage, and declined 230 again with the grain filling stage”. Is it common?

Explain this result is the discussion “These outcomes confirmed that by N splitting application and 237 delaying N quantity, a moderately higher Pn can be preserved at the later growth periods 238 of winter wheat and that the flowering stage is the best stage fertilizer application as top- 239 dressing.” The physiology of plant also influences on results?

This is the great result of study “Maximum NUE (0.15 and 0.11) was noted where 280 150 kg N ha-1 and 50% + 50% ratio was practiced and minimum N use efficiency (0.09 281 and 0.10) was observed where no N fertilizer was supplied at jointing stage as compared 282 to flowering and grain filling stage”.

In the Table, check the letter of NUE. I think that is wrong for the number “0.08±0.00b”. Please, check it.

Table 4, edit it. The quality is poor.

All text must be checked because there are some problems of grammar and editions.

In the conclusion, the authors could explore more physiologic results and explore the results. The conclusion is very poor

Author Response

Abstract: add an initial phrase of introduction.

Response: Thank you ver much for your time to improve our work. We have modified the abstract.

Explain the N rates in the abstract; what is the means for 5:5?

We have explained N rates in abstract part. The 5:5 means (50%: 50% N ratios) split application of nitrogen doses i.e. 50% at sowing time and 50% at jointing, 50% at flowering, and 50% at grain filling stage.

Line31: Check it “N225 kg ha-1”. What is N225?

N225 is nitrogen rate of 225 kg ha-1 applied to winter wheat as a treatment. We have modified in the abstract section.

The abstract could have more information about the experiment. Explore more details about the site.

We have modified the abstract.

Line 38: race crop?

It is a typo error, correct one is rice crop and modified in the manuscript.

In the Introduction could explore the N rates used commonly. There are few information about N management in wheat. Explore it.  

We have added the N rates and management practices in the introduction part.

Line82: what is LAI? NUE?

LAI (leaf area index) and NUE (nitrogen use efficiency). We have added in the manuscript.

The first paragraph in the Material and Methods is confused. Please, divide it.

The materials and methods section is improved significantly.

Soil characterization? Soil layer? Nutrient contents? Citation of soil methodologies used in characterization. How many samples?

Composite soil samples were collected from 0-20 cm depth. In the laboratory, soil samples were air dried and then passed from 2 mm sieve size. Later, some of the soil's fundamental chemical-physical characteristics were measured. Therefore, overall 30 samples were collected from different sampling points. Methods of measurements with references were added into manuscript.

Explain how N was applied in soil.

Nitrogen was applied through topdressing manually at various developmental stages throughout the experiment in the soil.

Line 108-126: too long. Please, divide it.

We have modified the sentences.

Is there lime application? Yes, rate? When? Or Why not?

There is no lime application in this experiment, only nitrogen fertilizer (source: urea 46.4%) was applied in the soil. SInce our main focus was to improve the nitrogen use effeciency and yield of winter wheat by using nitrogen fertilizers, because nitrogen is an imporant macronutrient for plant growth and development. So, by reshcduling and optimizing the nitrogen fertilizers, the yield of crop can be improved.

Line 200: What is AGB?

AGB represents the abve groung biomass, and we have modified in the manuscript.

The results of dry mass and height were not expected in the various developmental stages? Explain the results clearly because it is expected higher development with older plants.

The results of dry biomass and plant height was explained to compare the effects of different nitrogen doses at various development stages of winter wheat. So, it could be recommended to farmers at which stage different nitrogen rates and doses showed significant results and ultimately enhance the grain yield. Our study also optimized the nitrogen ratio and doses for better nitrogen efficiency and yield. The results are modified in manuscript, too.

Line 229: what is Pn?

Pn shows the photosynthetic rate and modified in the manuscript.

Line 230-231: “Pn improved up to the flowering stage, and declined 230 again with the grain filling stage”. Is it common?

Yes, it is a common phenomenon because at grain filling stage crop switched to physiological maturity and the process of photosynthesis declined as compared to flowering stage.

Explain this result is the discussion “These outcomes confirmed that by N splitting application and 237 delaying N quantity, a moderately higher Pn can be preserved at the later growth periods 238 of winter wheat and that the flowering stage is the best stage fertilizer application as top- 239 dressing.” The physiology of plant also influences on results?

We have explained these results in discussion section. As physiology of plants including photosynthesis process significantly influenced the results under different nitrogen doses.

This is the great result of study line 280, 281, and 282 “Maximum NUE (0.15 and 0.11) was noted where 150 kg N ha-1 and 50% + 50% ratio was practiced and minimum N use efficiency (0.09 and 0.10) was observed where no N fertilizer was supplied at jointing stage as compared to flowering and grain filling stage”.

Thank you for your encouragement and valuable comments.

In the Table, check the letter of NUE. I think that is wrong for the number “0.08±0.00b”. Please, check it.

We have modified value and lettering in the table.

Table 4, edit it. The quality is poor.

We have edited the table.

All text must be checked because there are some problems of grammar and editions.

Thank you for your valuable comments. We have revised this manuscript according to your suggestions.

In the conclusion, the authors could explore more physiologic results and explore the results. The conclusion is very poor

We have modified the conclusion part according to your suggestions.

Reviewer 3 Report

Lines 2-4: Title

The title is long and so I suggest the following title:

Improving winter wheat photosynthesis, nitrogen use efficiency, and yield by optimizing nitrogen fertilization

Line 26: ….of winter wheat (),..

Please provide the full scientific name of wheat at the first mention.

Line 27: … two N ratios (5:5 and 6:4)…

What are these rations? Ration of what to what?

Lines 26-28: The treatments, especially for time and ratio are ambiguous. Did you use N as splits? Ratios 5:5 or 6:4 are not clear, do you mean 50%-50% and 60%-40%?

Line 34: …with 60% + 40% at jointing stage increased …

Not clear! 60% for jointing stage, and 40% for?

Lines 127 and …: 2.3. Measurements

I suggest remove sub-sub headings (e.g. 2-3-1 to 2-3-7).

No need to present all detail on measurements, just mention the method and refer to a valid reference.

Short the sub-heading in results and discussion sections.

Use just one statistical method, standard error (SE) or letters (LSD). In some case these two statistical methods are contradictory.

Please mention which method was used for comparison of means (LSD) in caption of the figures.

Author Response

Lines 2-4: Title

The title is long and so I suggest the following title:

Improving winter wheat photosynthesis, nitrogen use efficiency, and yield by optimizing nitrogen fertilization

Response: Thank you for your sincere efforts in impring our work. We have modified the title accordingly.

Line 26: ….of winter wheat (),..

Please provide the full scientific name of wheat at the first mention.

We have write the full scientific name of wheat.

Line 27: … two N ratios (5:5 and 6:4)…

What are these rations? Ration of what to what?

It is 50%-50% and 60%-40% ratios of different nitrogen doses which are applied at various growth stages in split applications. We have added this details in the manuscript clearly.

Lines 26-28: The treatments, especially for time and ratio are ambiguous. Did you use N as splits? Ratios 5:5 or 6:4 are not clear, do you mean 50%-50% and 60%-40%?

Yes, it is  50%-50% and 60%-40% ratios of different nitrogen doses which are applied at various growth stages in split applications. In 50% + 50% ratio we applied nitrogen as a (50% at sowing time and 50% at jointing stage, 50% at sowing time + 50% at flowering stage time, 50% at sowing time + 50% at grain filling stage) and in 60% + 40% ratio we applied nitrogen as a (60% at sowing time and 40% at jointing stage, 60% at sowing time + 40% at flowering stage time, 60% at sowing time + 40% at grain filling stage.

Line 34: …with 60% + 40% at jointing stage increased …

Not clear! 60% for jointing stage, and 40% for?

It is 60% at sowing time and 40% at jointing time. We have clearly mentioned in the manuscript.

Lines 127 and …: 2.3. Measurements

I suggest remove sub-sub headings (e.g. 2-3-1 to 2-3-7).

We have removed the sub-headings.

No need to present all detail on measurements, just mention the method and refer to a valid reference.

We have mentioned the reference of all the reported methods.

Short the sub-heading in results and discussion sections.

We have revised the sub-headings of both sections results and discussion.

Use just one statistical method, standard error (SE) or letters (LSD). In some case these two statistical methods are contradictory.

We have modified the statical method throughout the figures and tables by standerizing the ±SE (standard error).

Please mention which method was used for comparison of means (LSD) in caption of the figures.

We have mentioned the comparsion method ±SE (standard error) in the caption of figures.

Round 2

Reviewer 1 Report

The authors have revised as requested and I have no further suggestions for the manuscript to be published.

Reviewer 2 Report

Accept in present form

Reviewer 3 Report

The authors corrected the manuscript based on the comments. So, I think it can be accepted after some correction in:

1. English improvement

2. The journal instruction